# SPATIAL DEPENDENCY NETWORKS: NEURAL LAYERS FOR IMPROVED GENERATIVE IMAGE MODELING

**Đorđe Miladinović** *
ETH Zurich
Zürich, Switzerland

**Aleksandar Stanić**
Swiss AI Lab IDSIA, USI
Lugano, Switzerland

**Stefan Bauer**
Max-Planck Institute
Tübingen, Germany

**Jürgen Schmidhuber**
Swiss AI Lab IDSIA, USI
Lugano, Switzerland

**Joachim M. Buhmann**
ETH Zurich
Zürich, Switzerland

## ABSTRACT

How to improve generative modeling by better exploiting spatial regularities and coherence in images? We introduce a novel neural network for building image generators (decoders) and apply it to variational autoencoders (VAEs). In our spatial dependency networks (SDNs), feature maps at each level of a deep neural net are computed in a spatially coherent way, using a sequential gating-based mechanism that distributes contextual information across 2-D space. We show that augmenting the decoder of a hierarchical VAE by spatial dependency layers considerably improves density estimation over baseline convolutional architectures and the state-of-the-art among the models within the same class. Furthermore, we demonstrate that SDN can be applied to large images by synthesizing samples of high quality and coherence. In a vanilla VAE setting, we find that a powerful SDN decoder also improves learning disentangled representations, indicating that neural architectures play an important role in this task. Our results suggest favoring spatial dependency over convolutional layers in various VAE settings. The accompanying source code is given at: `https://github.com/djordjemila/sdn`.

## 1 INTRODUCTION

The abundance of data and computation are often identified as core facilitators of the deep learning revolution. In addition to this technological leap, historically speaking, most major algorithmic advancements critically hinged on the existence of inductive biases, incorporating prior knowledge in different ways. Main breakthroughs in image recognition (Cireşan et al., 2012; Krizhevsky et al., 2012) were preceded by the long-standing pursuit for shift-invariant pattern recognition (Fukushima & Miyake, 1982) which catalyzed the ideas of weight sharing and convolutions (Waibel, 1987; Le-Cun et al., 1989). Recurrent networks (exploiting temporal recurrence) and transformers (modeling the "attention" bias) revolutionized the field of natural language processing (Mikolov et al., 2011; Vaswani et al., 2017). Visual representation learning is also often based on priors e.g. independence of latent factors (Schmidhuber, 1992; Bengio et al., 2013) or invariance to input transformations (Becker & Hinton, 1992; Chen et al., 2020). Clearly, one promising strategy to move forward is to introduce more structure into learning algorithms, and more knowledge on the problems and data.

Along this line of thought, we explore a way to improve the architecture of deep neural networks that generate images, here referred to as (deep) image generators, by incorporating prior assumptions based on topological image structure. More specifically, we aim to integrate the priors on spatial dependencies in images. We would like to enforce these priors on all intermediate image representations produced by an image generator, including the last one from which the final image is synthesized. To that end, we introduce a class of neural networks designed specifically for building image generators – spatial dependency network (SDN). Concretely, spatial dependency layers of

---

*Correspondence at: djordjem@ethz.ch

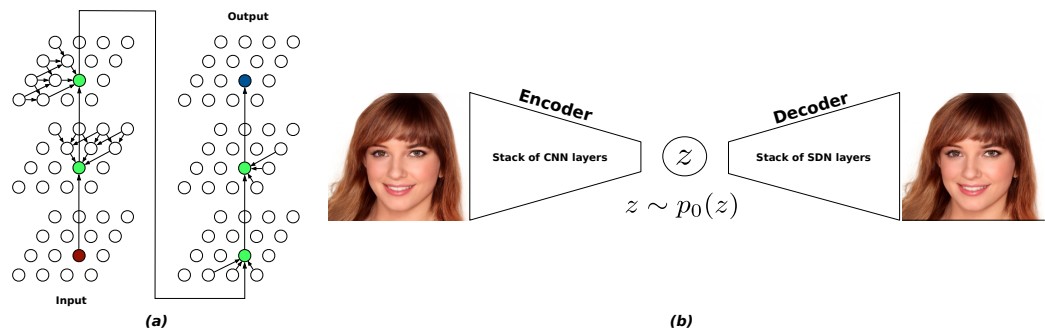

Figure 1: **(a) DAG of a spatial dependency layer.** The input feature vector (red node) is gradually refined (green nodes) as the computation progresses through the four sub-layers until the output feature vector is produced (blue node). In each sub-layer, the feature vector is corrected based on contextual information. Conditional maps within sub-layers are implemented as learnable deterministic functions with shared parameters; **(b) VAE with SDN decoder reconstructing a 'celebrity'.**

SDN (SDN layers) incorporate two priors: *(i)* spatial coherence reflects our assumption that feature vectors should be dependent on each other in a spatially consistent, smooth way. Thus in SDN, the neighboring feature vectors tend to be more similar than the non-neighboring ones, where the similarity correlates with the 2-D distance. The graphical model (Figure 1a) captures this assumption. Note also that due to their unbounded receptive field, SDN layers model long-range dependencies; *(ii)* spatial dependencies between feature vectors should not depend on their 2-D coordinates. Mathematically speaking, SDN should be equivariant to spatial translation, in the same way convolutional networks (CNN) are. This is achieved through parameter (weight) sharing both in SDN and CNN.

The particular focus of this paper is the application of SDN to variational autoencoders (VAEs) (Kingma & Welling, 2013). The main motivation is to improve the performance of VAE generative models by endowing their image decoders with spatial dependency layers (Figure 1b). While out of the scope of this work, SDN could also be applied to other generative models, e.g. generative adversarial networks (Goodfellow et al., 2014). More generally, SDN could be potentially used in any task in which image generation is required, such as image-to-image translation, super-resolution, image inpainting, image segmentation, or scene labeling.

SDN is experimentally examined in two different settings. In the context of real-life-image density modeling, SDN-empowered hierarchical VAE is shown to reach considerably higher test log-likelihoods than the baseline CNN-based architectures and can synthesize perceptually appealing and coherent images even at high sampling temperatures. In a synthetic data setting, we observe that enhancing a non-hierarchical VAE with an SDN facilitates learning of factorial latent codes, suggesting that unsupervised 'disentanglement' of representations can be bettered by using more powerful neural architectures, where SDN stands out as a good candidate model. The contributions and the contents of this paper can be summarized as follows.

CONTRIBUTIONS AND CONTENTS:

- The architecture of SDN is introduced and then contrasted to the related ones such as convolutional networks and self-attention.
- The architecture of SDN-VAE is introduced – the result of applying SDN to IAF-VAE (Kingma et al., 2016), with modest modifications to the original architecture. SDN-VAE is evaluated in terms of: *(a)* density estimation, where the performance is considerably improved both upon the baseline and related approaches to non-autoregressive modeling, on CIFAR10 and ImageNet32 data sets; *(b)* image synthesis, where images of competitively high perceptual quality are generated based on CelebAHQ256 data set.
- By integrating SDN into a relatively simple, non-hierarchical VAE, trained on the synthetic 3D-Shapes data set, we demonstrate in another comparative analysis substantial improvements upon convolutional networks in terms of: *(a)* optimization of the evidence lower bound (ELBO); *(b)* learning of disentangled representations with respect to two popular metrics, when $\beta$-VAE (Higgins et al., 2016) is used as an objective function.

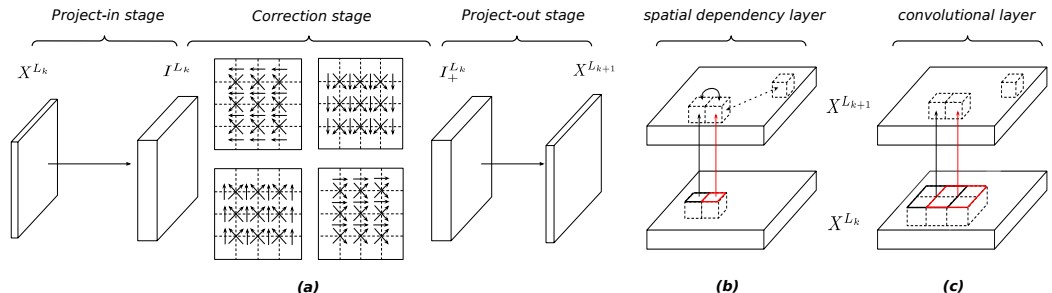

Figure 2: **Spatial dependency layer.** **(a)** a 3-stage pipeline; **(b)** dependencies between the input $X^{L_k}$ and the output $X^{L_{k+1}}$ modeled by a spatial dependency layer $L_k$. Solid arrows represent direct, and dashed indirect (non-neighboring) dependencies. Note that the sub-layers from Figure 1a are regarded as 'latent' here; **(c)** dependencies modeled by a $2 \times 2$ convolutional layer.

## 2 SDN ARCHITECTURE

The architectural design of SDN is motivated by our intent to improve modeling of spatial coherence and higher-order statistical correlations between pixels. By SDN, we refer to any neural network that contains at least one spatial dependency layer (Figure 2). Technically speaking, each layer of an SDN is a differentiable function that transforms input to output feature maps in a 3-stage procedure.

We first describe the SDN layer in detail, then address practical and computational concerns.

**Project-in stage.** Using a simple feature vector-wise affine transformation, the input representation $X^{L_k}$ at the layer $L_k$ is transformed into the intermediate representation $I^{L_k}$:

$$I_{i,j}^{L_k} = X_{i,j}^{L_k} \mathbf{W^{(1)}} + \mathbf{b^{(1)}} \tag{1}$$

where $i$ and $j$ are the corresponding 2-D coordinates in $X^{L_k}$ and $I^{L_k}$, $\mathbf{b^{(1)}}$ is a vector of learnable biases and $\mathbf{W^{(1)}}$ is a learnable matrix of weights whose dimensions depend on the number of channels of $X^{L_k}$ and $I^{L_k}$. In the absence of overfitting, it is advisable to increase the number of channels to enhance the memory capacity of $I^{L_k}$ and enable unimpeded information flow in the correction stage. Notice that described transformation corresponds to the convolutional (CNN) operator with a kernel of size 1. If the scales between the input $X^{L_k}$ and the output $X^{L_{k+1}}$ differ (e.g. if converting an 8x8 to a 16x16 tensor), in this stage we additionally perform upsampling by applying the corresponding transposed CNN operator, also known as 'deconvolution' (Zeiler & Fergus, 2014).

**Correction stage.** The elements of $I^{L_k}$ are 'corrected' in a 4-step procedure. In each step, a unidirectional sweep through $I^{L_k}$ is performed in one of the 4 possible directions: *(i)* bottom-to-top; *(ii)* right-to-left; *(iii)* left-to-right; and *(iv)* top-to-bottom. During a single sweep, the elements of the particular column/row are computed in parallel. The corresponding correction equations implement controlled updates of $I^{L_k}$ using a gating mechanism, as done in popular recurrent units (Hochreiter & Schmidhuber, 1997; Cho et al., 2014). Specifically, we use a multiplicative update akin to the one found in GRUs (Cho et al., 2014), but adapted to the spatial setting. The procedure for the bottom-to-top sweep is given in Algorithm 1 where $\mathbf{W}^*$ denote learnable weights, $\mathbf{B}^*$ are learnable biases, $i$ is the height (growing bottom-to-top), $j$ is the width and $c$ is the channel dimension. Gating elements $r_{i,j,c}$ and $z_{i,j,c}$ control the correction contributions of the *a priori* value of $I_{i,j,c}$ and the proposed value $n_{i,j,c}$, respectively. For the borderline elements of $I^{L_k}$, missing neighbors are initialized to zeros (hence the zero padding). The sweeps in the remaining 3 directions are performed analogously.

**Project-out stage.** Corrected (*a posteriori*) representation $I_+^{L_k}$ is then mapped to the output $X^{L_{k+1}}$, which has the same number of channels as the input $X^{L_k}$:

$$X_{i,j}^{L_{k+1}} = I_{+,i,j}^{L_k} \mathbf{W^{(3)}} + \mathbf{b^{(3)}} \tag{2}$$

where $\mathbf{W^{(3)}}$ is the stage-level learnable weight matrix and $\mathbf{b^{(3)}}$ is the corresponding bias vector.

---

**Algorithm 1** Bottom-to-top sweep of the correction stage

---

**Input:** $I^{L_k}$ – intermediate representation; $N$ – the scale of $I^{L_k}$;
**Output:** $I^{L_k}_+$ – corrected intermediate representation; # after all 4 sweeps
**Complexity:** $\mathcal{O}(N)$
$I^{L_k} = \tanh(I^{L_k})$ # done for the first direction only
$I^{L_k} = zero\_padding(I^{L_k})$ # done for the first direction only
**for** $i = 1$ **to** $N$ and ($j = 1$ **to** $N$ **in parallel**) **do**
  $r_{i,j,c} = \sigma([\mathbf{W^{r1}}I_{i,j}]_c + [\mathbf{W^{r2}}I_{i-1,j}]_c + [\mathbf{W^{r3}}I_{i-1,j-1}]_c) + [\mathbf{W^{r4}}I_{i-1,j+1}]_c + \mathbf{B^r}_{i,j,c})$ # reset gate
  $z_{i,j,c} = \sigma([\mathbf{W^{z1}}I_{i,j}]_c + [\mathbf{W^{z2}}I_{i-1,j}]_c + [\mathbf{W^{z3}}I_{i-1,j-1}]_c) + [\mathbf{W^{z4}}I_{i-1,j+1}]_c + \mathbf{B^z}_{i,j,c})$ #update gate

  $n_{i,j,c} = \tanh(r_{i,j,c}[\mathbf{W^{n1}}I_{i,j}]_c + [\mathbf{W^{n2}}I_{i-1,j}]_c + [\mathbf{W^{n3}}I_{i-1,j-1}]_c) + [\mathbf{W^{n4}}I_{i-1,j+1}]_c + \mathbf{B^n}_{i,j,c})$
  $I_{i,j,c} = z_{i,j,c}I_{i,j,c} + (1 - z_{i,j,c})n_{i,j,c}$ # feature correction
**end for**

---

**Computational considerations.** A valid concern in terms of scaling SDN to large images is the computational complexity of $\mathcal{O}(N)$, $N$ being the scale of the output. We make two key observations:

a. one can operate with less than $4$ directions per layer, alternating them across layers to achieve the 'mixing' effect. This improves runtime with some sacrifice in performance;

b. in practice, we found it sufficient to apply spatial dependency layers only at lower scales (e.g. up to 64x64 on CelebAHQ256), with little to no loss in performance. We speculate that this is due to redundancies in high-resolution images, which allows modelling of relevant spatial structure at lower scales;

To provide additional insights on the computational characteristics, we compared SDN to CNN in a unit test-like setting (Appendix A.3). Our implementation of a 2-directional SDN has roughly the same number of parameters as a 5×5 CNN, and is around 10 times slower than a 3×3 CNN. However, we also observed that in a VAE context, the overall execution time discrepancy is (about 2-3 times) smaller, partially because SDN converges in fewer iterations (Table 4 in Appendix).

**Avoiding vanishing gradients.** The problem might arise when SDN layers are plugged in a very deep neural network (such as the one discussed in Section 4). To remedy this, in our SDN-VAE experiments we used the variant of an SDN layer with a 'highway' connection, i.e. gated residual (Srivastava et al., 2015), as shown in Figure 3.

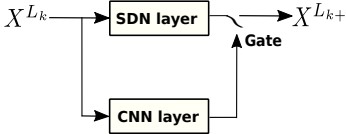

Figure 3: **Residual SDN layer.**

## 3 RELATED WORK

**Convolutional networks.** Both spatial dependency and convolutional layers exploit locality (a feature vector is corrected based on its direct neighbors) and equivariance to translation (achieved through parameter sharing). Additionally, SDN enforces spatial coherence. There are also differences in the dependency modeling (depicted in Figures 2b and 2c). Firstly, when conditioned on the input $X^{L_k}$, the output feature vectors of a convolutional layer are statistically independent: $X^{L_{k+1}}_{i,j} \perp\!\!\!\perp X^{L_{k+1}}_{l,m}|X^{L_k}\ \forall(i,j) \neq (l,m)$, where $i$ and $j$ are feature map coordinates. In spatial dependency layers, all feature vectors are dependent: $X^{L_{k+1}}_{i,j} \not\perp\!\!\!\perp X^{L_{k+1}}_{l,m}|X^{L_k}\ \forall(i,j) \neq (l,m)$. Secondly, the unconditional dependency neighborhood of a convolutional layer is bounded by the size of its receptive field – a conceptual limitation not present in spatial dependency layers. Hence, spatial dependency layers can model long-range spatial dependencies.

**Autoregressive models and normalizing flows.** SDN design is inspired by how autoregressive models capture spatial dependencies due to the pixel-by-pixel generation process (Theis & Bethge, 2015). SDN(-VAE) improves on this is by modeling spatial dependencies in multiple directions. Autoregressive models are inherently uni-directional; the generation order of pixels can be changed but is fixed for training and sampling. Thus in some sense, SDN transfers the ideas from autoregressive to non-autoregressive settings in which there are no ordering constraints. Also, most autoregressive

models use teacher forcing during training to avoid sequential computation, but sampling time complexity is quadratic. SDN has linear complexity in both cases and can operate at smaller scales only. Parallel computation of SDN is similar to the one found in PixelRNN (Van Oord et al., 2016), but instantiated in the non-autoregressive setting with no directionality or conditioning constraints.

One can also draw parallels with how autoregressive models describe normalizing flows, for example, IAF (Kingma et al., 2016) and MAF (Papamakarios et al., 2017). In this case, each flow in a stack of flows is described with an autoregressive model that operates in a different direction, or more generally, has a different output ordering. In its standard 4-directional form (Figure 1(a)), SDN creates dependencies between all input and output feature vectors; this renders a full, difficult-to-handle Jacobian matrix. For this reason, SDN is not directly suitable for parameterizing normalizing flows in the same way. In contrast, the main application domain of SDN is rather the parameterization of deterministic, unconstrained mappings.

**Self-attention.** The attention mechanism has been one of the most prominent models in the domain of text generation (Vaswani et al., 2017). Recently, it has also been applied to generative image modeling (Parmar et al., 2018; Zhang et al., 2019). Both SDN and self-attention can model long-range dependencies. The key difference is how this is done. In SDN, only neighboring feature vectors are dependent on each other (see Figures 1 and 2) hence the spatial locality is exploited. Gated units are used to ensure that the information is propagated across large distances. On the other hand, self-attention requires an additional mechanism to incorporate positional information, but the dependencies between non-neighboring feature vectors are direct. We believe that the two models should be treated as complementary; self-attention excels in capturing long-range dependencies while SDN is better in enforcing spatial coherence, equivariance to translation, and locality. Note finally that standard self-attention is costly, with quadratic complexity in time and space in the number of pixels, which makes it $\mathcal{O}(N^4)$ in the feature map scale, if implemented naively.

**Other related work.** SDNs are also notably reminiscent of Multi-Dimensional RNNs (Graves et al., 2007) which were used for image segmentation (Graves et al., 2007; Stollenga et al., 2015) and recognition (Visin et al., 2015), but not in the context of generative modeling. One technical advantage of our work is the linear training time complexity in the feature map scale. Another difference is that SDN uses GRU, which is less expressive than LSTM, but more memory efficient.

## 4 SDN-VAE: AN IMPROVED VARIATIONAL AUTOENCODER FOR IMAGES

As a first use case, SDN is used in the SDN-VAE architecture. The basis of SDN-VAE is the IAF-VAE model (Kingma et al., 2016). Apart from integrating spatial dependency layers, we introduce additional modifications for improved performance, training stability, and reduced memory requirements. We first cover the basics of VAEs, then briefly summarize IAF-VAE and the related work. Finally, we elaborate on the novelties in the SDN-VAE design.

**Background on VAEs.** VAEs assume a latent variable generative model $p_\theta(X) = \int p_\theta(X, Z)dZ$ where $\theta$ are model parameters. When framed in the maximum likelihood setting, the marginal probability of data is intractable due to the integral in the above expression. VAEs take a variational approach, approximating posterior $q_\phi(Z|X)$ using a learnable function $q$ with parameters $\phi$. Following Kingma & Welling (2019), we can derive the following equality:

$$\log p_\theta(X) = \underbrace{\mathbb{E}_{q_\phi(Z|X)}\left[\log\left[\frac{p_\theta(X, Z)}{q_\phi(Z|X)}\right]\right]}_{\mathcal{L}_{\phi,\theta}(X)=ELBO} + \underbrace{\mathbb{E}_{q_\phi(Z|X)}\left[\log\left[\frac{q_\phi(Z|X)}{p_\theta(Z|X)}\right]\right]}_{KL(q_\phi(Z|X)||p_\theta(Z|X))\geq 0} \quad (3)$$

$\mathcal{L}_{\phi,\theta}(X)$ is the evidence lower bound (ELBO), since it 'bounds' the marginal log-likelihood term $\log p_\theta(X)$, where the gap is defined by the KL-divergence term on the right. In VAEs, both $q_\phi$ and $p_\theta$ are parametrized by deep neural networks. The model is learned via backpropagation, and a low variance approximation of the gradient $\nabla_{\phi,\theta}\mathcal{L}_{\phi,\theta}(X)$ can be computed via the *reparametrization trick* (Kingma & Welling, 2019; Rezende et al., 2014).

**Background on IAF-VAE.** Main advancements of IAF-VAE in comparison to the vanilla VAE include: *(a)* leveraging on a hierarchical (ladder) network of stochastic variables (Sønderby et al.,

2016) for increased expressiveness and improved training; *(b)* introducing inverse autoregressive flows to enrich simple isotropic Gaussian-based posterior with a normalizing flow; *(c)* utilizing a structured bi-directional inference procedure to encode latent variables through an encoder-decoder interplay; *(d)* introducing a ResNet block as a residual-based (Srivastava et al., 2015; He et al., 2016) version of the ladder layer. For more details, please see the original paper (Kingma et al., 2016).

**Related work.** BIVA (Maaløe et al., 2019) is an extension of IAF-VAE which adds skip connections across latent layers, and a stochastic bottom-up path. In our experiments, however, we were not able to improve performance in a similar way. Concurrently to our work, NVAE (Vahdat & Kautz, 2020) reported significant improvements upon the IAF-VAE baseline, by leveraging on many technical advancements including: batch normalization, depth-wise separable convolutions, mixed-precision, spectral normalization and residual parameterization of the approximate posterior. Both methods use the discretized mixture of logistics (Salimans et al., 2017).

## SDN-VAE ARCHITECTURE

Our main contribution to the IAF-VAE architecture is the integration of residual spatial dependency layers, or ResSDN layers (Figure 3). We replace the convolutional layers on the top-down (generation) path of the IAF-VAE ResNet block, up to a particular scale: up to 32x32 for images of that resolution, and up to 64x64 for 256x256 images. For the convenience of the reader, we borrowed the scheme from the original paper and clearly marked the modifications (Figure 4).

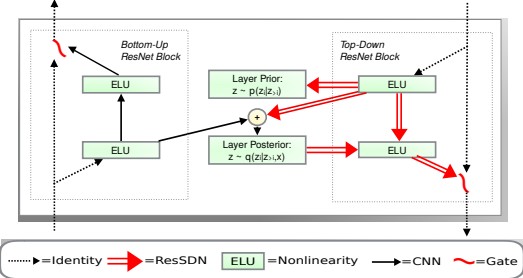

Figure 4: **SDN-VAE ResNet block as modified IAF-VAE ResNet block**. Two notable changes (marked in red) include applying: *(a)* ResSDN layers instead of convolutional layers on the top-down path; *(b)* gated rather than a sum residual; (figure adapted from Kingma et al. (2016))

Other notable modifications include: *(a)* gated residual instead of a sum residual in the ResNet block (Figure 4), for improved training stability; *(b)* more flexible observation model based on a discretized mixture of logistics instead of a discretized logistic distribution, for improved performance; and *(c)* mixed-precision (Micikevicius et al., 2018), which reduced memory requirements thus allowing training with larger batches. Without ResSDN layers, we refer to this architecture as IAF-VAE+.

## 5 IMAGE DENSITY MODELING

Proposed SDN-VAE and the contributions of spatial dependency layers to its performance were empirically evaluated in terms of: *(a)* density estimation, where SDN-VAE was in a quantitative fashion compared to the convolutional network baseline, and to the related state-of-the-art approaches; *(b)* image synthesis, where a learned distribution was analyzed in a qualitative way by sampling from it in different ways. Exact details of our experiments, additional ablation studies and additional results in image synthesis are documented in the Appendix sections A.1, A.2 and A.4 respectively.

**Density estimation.** From a set of i.i.d. images $\mathcal{D}_{train} = X_{1..N}$, the true probability density function $p(X)$ is estimated via a parametric model $p_\theta(X)$, whose parameters $\theta$ are learned using the maximum log-likelihood objective: $\arg\max_\theta \left[ \log p_\theta(X) \approx \frac{1}{N} \sum_{i=1}^{N} \log p_\theta(X_i) \right]$. The test log-likelihood is computed on an isolated set of images $\mathcal{D}_{test} = X_{1..K}$, to evaluate learned $p_\theta(X)$.

SDN-VAE and the competing methods were tested on CIFAR-10 (Krizhevsky et al., 2009), ImageNet32 (Van Oord et al., 2016), and CelebAHQ256 (Karras et al., 2017). Quantitative comparison is given in Table 1. IAF-VAE+ denotes our technically enhanced implementation of IAF-VAE, but without the SDN modules (recall Section 4).

| Type | Method | CIFAR-10 | ImageNet32 | CelebAHQ256 |
|------|--------|----------|------------|-------------|
| **VAE-based** | **SDN-VAE (ours)** | (2.87) | (3.85) | (0.70) |
| | IAF-VAE+ (ours) | 3.05 | 4.00 | 0.71 |
| | IAF-VAE (Kingma et al., 2016) | 3.11 | X | X |
| | BIVA (Maaløe et al., 2019) | 3.08 | X | X |
| | NVAE (Vahdat & Kautz, 2020) | 2.91 | 3.92 | (0.70) |
| **Flow-based** | GLOW (Kingma & Dhariwal, 2018) | 3.35 | 4.09 | 1.03 |
| | FLOW++ (Ho et al., 2019) | 3.08 | 3.86 | X |
| | ANF (Huang et al., 2020) | 3.05 | 3.92 | 0.72 |
| | SurVAE (Nielsen et al., 2020) | 3.08 | 4.00 | X |
| **Autoregressive*** | PixelRNN (Van Oord et al., 2016) | 3.00 | 3.86 | X |
| | PixelCNN (Van den Oord et al., 2016) | 3.03 | 3.83 | X |
| | PixelCNN++ (Salimans et al., 2017) | 2.92 | X | X |
| | PixelSNAIL (Chen et al., 2018) | **2.85** | 3.80 | X |
| | SPN (Menick & Kalchbrenner, 2018) | X | 3.85 | **0.61** |
| | IT (Parmar et al., 2018) | 2.90 | **3.77** | X |

Table 1: **Density estimation results.** Negative test log-likelihood is measured in *bits per dimension (BPD)* - lower is better. As in previous works, only the most successful runs are reported. Circled are the best runs among non-autoregressive models and bolded are the best runs overall.
* by autoregressive we refer to the methods based on $p(X) = \prod_{i=1} p(X_i|X_1, ..., X_{i-1})$ factorization.

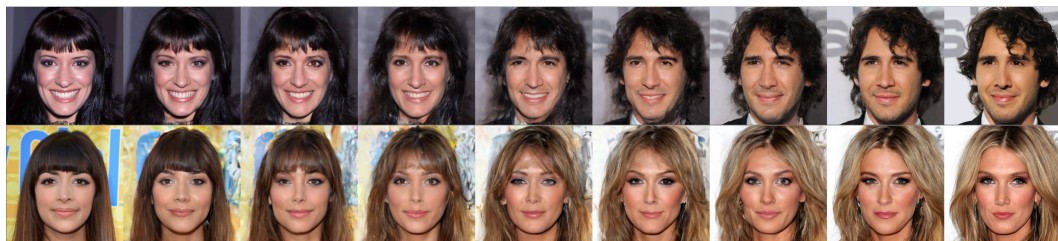

Figure 5: **Linear interpolation between test images.** Given a pair of images from the test dataset (on the far right and far left), linear interpolation was performed on the layer 5 at a temperature of 0.9 (the sequence above) and on the layer 4 at a temperature of 1.0 (the sequence below).

**Image synthesis.** We additionally inspected the trained $p_\theta(X)$ by: *(a)* unconditionally generating samples from the prior, at different temperatures (Figure 6 left); *(b)* sampling in the neighborhood of an image not seen during training time (Figure 6 right); *(c)* interpolating between two images not seen during training time (Figure 5). All technical details on these experiments are in Appendix A.1. Additional images are given in Appendix A.4.

**Discussion.** Our results clearly suggest that SDN-VAE is not only superior to the convolutional network baseline, but also to all previous related approaches for non-autoregressive density modeling. To further inspect whether the increase in performance is due to applying SDN or due to the implicit increase in the number of parameters, we conducted additional ablation studies (described in Appendix A.2). These results provide additional evidence that SDN is the cause of increased

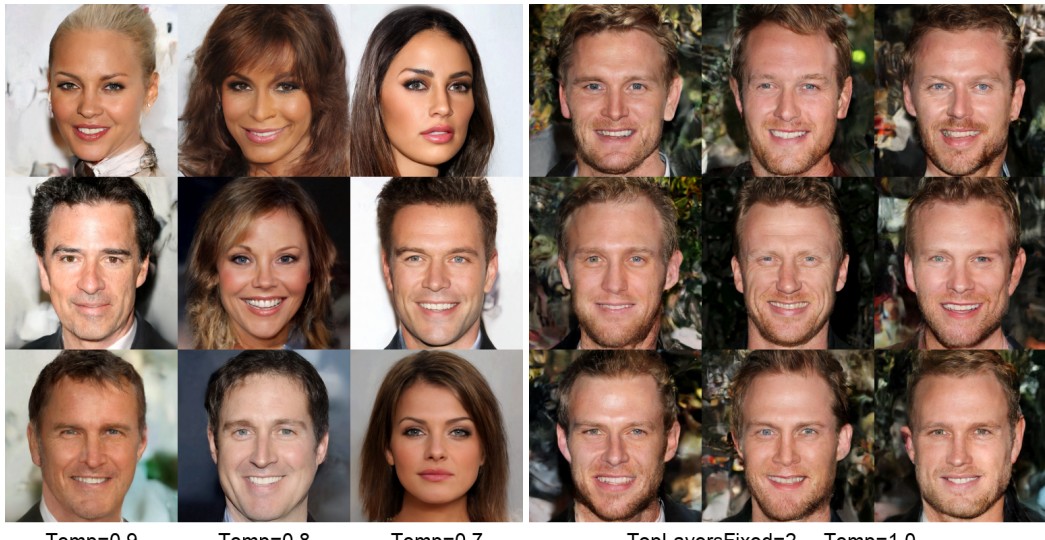

Temp=0.9     Temp=0.8     Temp=0.7        TopLayersFixed=2    Temp=1.0

Figure 6: *(left)* **Unconditional sampling.** Samples were drawn from the SDN-VAE prior at varying temperatures; *(right)* **Sampling around a test image.** Conditioned on the top 2 layers of the latent code of the image in the center, the surrounding images were sampled at a temperature of 1.0.

performance, as even convolutional networks with more parameters, increased receptive field or more depth lead to inferior performance. In our image generation experiments we confirmed that SDN-VAE can be successfully scaled to large images. Generated samples are of exceptionally high quality. What is particularly challenging in VAE-based image generation, is sampling at high temperatures, however as shown in Figure 10 of our Appendix, SDN-VAE can successfully synthesize coherent images even in this case.

# 6 LEARNING DISENTANGLED REPRESENTATIONS

To study its performance impact in a more constrained setting, SDN was paired with a VAE architecturally much simpler than IAF-VAE. Apart from the implementation simplicity and shorter training time, non-hierarchical VAE is more suitable for disentangled representation learning, at least in the sense of (Higgins et al., 2016) where the aim is to decorrelate the dimensions of a latent vector. In particular, the gains in performance when using SDN were evaluated with respect to: *(a)* evidence lower bound (ELBO); *(b)* disentanglement of latent codes based on the corresponding metrics, to examine the effects of SDN decoder to the quality of learned latent representations.

**Experiment setting.** A relatively simple VAE architecture with a stack of convolutional layers in both the encoder and decoder (Higgins et al., 2016) was used as a baseline model. SDN decoder was constructed using a single non-residual, one-directional instance of spatial dependency layer placed at the scale 32. The competing methods were evaluated on 3D-Shapes (Burgess & Kim, 2018), a synthetic dataset containing $64 \times 64$ images of scenes with rooms and objects of various colors and shapes. Following related literature (Locatello et al., 2019), there was no training-test split, so the reported ELBOs are measured on the training set. The remaining details are in Appendix A.5.

**Training with an unmodified VAE objective.** Using the original VAE objective (Kingma & Welling, 2013), competing architectures were compared in terms of ELBO (Table 2). SDN leads to substantially better estimates which come from better approximation of the conditional log-likelihood term, as shown in more detail in Figure 12 in the Appendix A.6.

| VAE | KLD [BPD$\times 10^{-3}$] | Neg. ELBO [BPD] |
|-----|---------------------------|-----------------|
| CNN | $7.40 \pm 0.07$ | $4.44 \pm 0.04$ |
| SDN | $7.95 \pm 0.07$ | $\mathbf{2.66 \pm 0.04}$ |

Table 2: **Density estimation on 3D-Shapes.**

**Training with $\beta$-VAE objective.** The same models were trained using $\beta$-VAE objective (Higgins et al., 2016). In the $\beta$-VAE algorithm, the 'disentanglement' of latent dimensions is controlled by an additional hyperparameter denoted as $\beta$, which effectively acts as a regularizer. Modified ELBO reads as: $\mathcal{L}_{\phi,\theta}(X, \beta) = \mathbb{E}_{q_\phi(Z|X)}[\log p_\theta(X|Z)] - \beta \, \mathrm{KL}(q_\phi(Z|X)||p_0(Z))$. To investigate the effects of SDN decoder on the shaping of the latent space, we measured disentanglement for different values of $\beta$, for two popular disentanglement metrics: $\beta$-VAE (Higgins et al., 2016) and FactorVAE (Kim & Mnih, 2018). The results are shown in Figure 7.

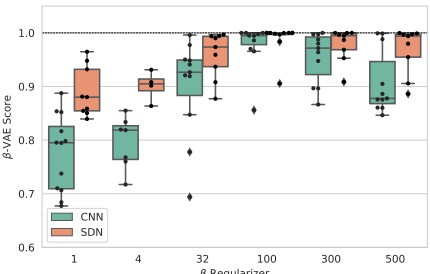 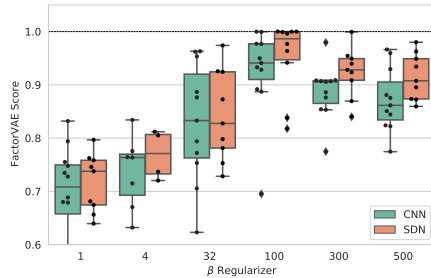

Figure 7: **$\beta$-VAE disentanglement results.** The plots compare the CNN and SDN-based VAE architectures, with respect to the $\beta$-VAE (left) and FactorVAE (right) disentanglement metrics.

**Discussion.** In a more constrained VAE setting, the beneficial effects of SDN to image modeling were confirmed. Perhaps surprisingly at first glance, we also found that improved VAE decoder facilitates factorization of latent codes. This may seem contradictory to the previous findings (Bowman et al., 2015; Chen et al., 2016) which suggest that a powerful decoder hurts representation learning. However, the difference here is that SDN decoders do not leverage on teacher forcing. 'Posterior collapse' can happen only very early in the training, but this is in our case easily solved using $\beta$ annealing (warm-up) procedure (Bowman et al., 2015). Our results indicate the importance of good neural architectures in learning disentangled representations.

## 7 CONCLUSIONS

*Summary.* This paper introduced novel spatial dependency layers suitable for deep neural networks that produce images – image generators. The proposed SDN is tailored to image generators that operate in a non-autoregressive way, i.e. synthesize all pixels 'at once'. Spatial dependency layers improve upon convolutional layers by modeling spatial coherence and long-range spatial dependencies. The main motivation behind introducing SDN is its application to VAE image decoders. SDN was analyzed in the context of: *(a)* a complex hierarchical VAE model, where the state-of-the-art performance was obtained in non-autoregressive density modeling; *(b)* a vanilla VAE, resulting in improvements in both density modeling and disentangled representation learning.

*Implications.* Spatial dependency layer is a simple-to-use module that can be easily integrated into any deep neural network, and as demonstrated in the paper, it is favorable to a convolutional layer in multiple settings. SDN was shown to improve the performance of VAEs immensely, which is relevant since VAEs can be used in any density modeling task (unlike generative adversarial networks), e.g. outlier detection. VAEs are also favorable to autoregressive models in the settings where an explicit latent representation of data is needed, e.g. when the latent interpolation between test samples is necessary. We also provide an insight how VAE decoders can be changed for improved representation learning, suggesting an alternative way, and a concrete solution for this problem.

*Limitations and future work.* The main downside of SDN remains the computation time. However, we suspect that a more optimized implementation could substantially improve the runtime performance of SDN. In the future, it would be beneficial to explore the applicability of SDN in other settings. For example, one could apply SDN to other VAE architectures or to generative adversarial networks. SDN could also be applied to image processing tasks such as image super-resolution or image segmentation. It would also be interesting to explore the applicability of SDN for learning structured and disentangled representations of video sequences (Miladinović et al., 2019).

ACKNOWLEDGMENTS

We thank Max Paulus and Mihajlo Milenković for helpful comments and fruitful discussions. This research was supported by the Swiss National Science Foundation grant 407540_167278 EVAC - Employing Video Analytics for Crisis Management, and the Swiss National Science Foundation grant 200021_165675/1 (successor project: no: 200021_192356).

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

# A    APPENDIX

## A.1    DENSITY ESTIMATION – EXPERIMENT DETAILS

|  | CIFAR10 $32^2 = 32\text{x}32$ | ImageNet32 $32^2$ | CelebAHQ256 $256^2$ |
|---|---|---|---|
| # training samples | 50K | 1.281M | 27K |
| # test samples | 10K | 50K | 3K |
| Quantization | 8 bits | 8 bits | 5 bits |
| Encoder/decoder depth | 15 | 16 | 17 |
| Deterministic #channels per layer | 200 | 200 | 200 |
| Stochastic #channels per layer | 4 | 4 | 8 |
| Scales | $8^2, 16^2, 32^2$ | $4^2, 8^2, 16^2, 32^2$ | $4^2, 8^2, 16^2, 32^2, 64^2, 128^2, 256^2$ |
| SDN #channels per scale | 120, 240, 260 | 424 for all | 360, 360, 360, 360, 360, ~~SDN~~, ~~SDN~~ |
| Layers per scale | 5 for all | 4 for all | 2,2,2,2,2,3,4 |
| Number of directions per SDN | 2 | 3 | 2 |
| Optimizer | Adamax | Adamax | Adamax |
| Learning rate | 0.002 | 0.002 | 0.001 |
| Learning rate annealing | Exponential | Exponential | Exponential |
| Batch size per GPU | 32 | 32 | 4 |
| Number of GPUs | 8 | 8 | 8 |
| GPU type | TeslaV100 | TeslaV100 | TeslaV100 |
| GPU memory | 32GB | 32GB | 32GB |
| Prior model | Gaussian | Gaussian | Gaussian |
| Posterior model | Gaussian | Gaussian | Gaussian |
| Posterior flow | 1 IAF | 1 IAF | 1 IAF |
| Observation model | DML | DML | DML |
| DML Mixture components | 5 | 10 | 30 |
| Exponential Moving Average (EMA) | 0.999 | 0.9995 | 0.999 |
| Free bits | 0.01 | 0.01 | 0.01 |
| Number of importance samples | 1024 | 1024 | 1024 |
| Mixed-precision | Yes | Yes | Yes |
| Weight normalization | Yes | Yes | Yes |
| Horizontal flip data augmentation | Yes | No | Yes |
| Training time | 45h | 200h | 90h |

Table 3: **Experimental configurations for the density estimation tests.** DML is the discretized mixture of logistics (Salimans et al., 2017). Weight normalization, mixed-precision, free bits and Adamax are documented by Salimans & Kingma (2016); Micikevicius et al. (2018); Kingma et al. (2016); Kingma & Ba (2014) respectively. ~~SDN~~ means that SDN was not applied at the corresponding scale. The IAF (Kingma et al., 2016) contained 2 layers of masked 3x3 convolutional networks, with the context and number of CNN filters both of size 100. The importance sampling (Burda et al., 2015) was used for obtaining tighter lower bounds.

**On configuring hyperparameters.** Due to high computational demands, we only modestly optimized hyperparameters. In principle, our strategy was to scale up: (a) the number of deterministic and SDN filters; (b) batch size; (c) learning rate; (d) the number of spatial directions and (e) the number of DML components; as long as we found no signs of overfitting, had no memory or stability issues, and the training was not considerably slower. EMA coefficient values were taken from the previous related works (Kingma et al., 2016; Chen et al., 2018) – we tested 0.999 and 0.9995. We also swept through the values $\{2, 4, 8, 15\}$ for the number of latent stochastic filters, but saw no significant difference in the performance. Most extensively we explored the number of layers per scale, and found this to have relevant impact on runtime and over/underfitting, for both baseline and our architecture. We found that more downsampling improved runtime and reduced memory consumption, but made the tested models prone to overfitting.

## A.2 DENSITY ESTIMATION – ABLATION STUDIES

Additional ablation studies on SDN-VAE were conducted in order to explore in more detail whether the good performance on the density estimation experiments (Table 1) indeed comes from the proposed SDN layer. We used CIFAR-10 data on which both baseline and SDN-VAE models converged faster in comparison to two other data sets. Our main motivation was to understand whether the increase in performance comes from our architectural decisions or simply from the increase in the number of parameters. Namely, since the ResSDN layer (from Figure 3) is a network with a larger number of parameters in comparison to the baseline 3x3 convolutional layer, it would be justified to question if the increase in performance can be indeed attributed to ResSDN layers.

To that end, we designed multiple baseline blocks to replace convolutional layers in the IAF-VAE+ architecture, by replicating the design protocol from Section 4 in which ResSDN layer was applied to create the SDN-VAE architecture. Baseline blocks are illustrated in Figure 8. The main idea is to explore whether possibly increased kernel of a CNN would be sufficient to model spatial dependencies, or whether a deeper network can bring the same performance to the basic VAE architecture.

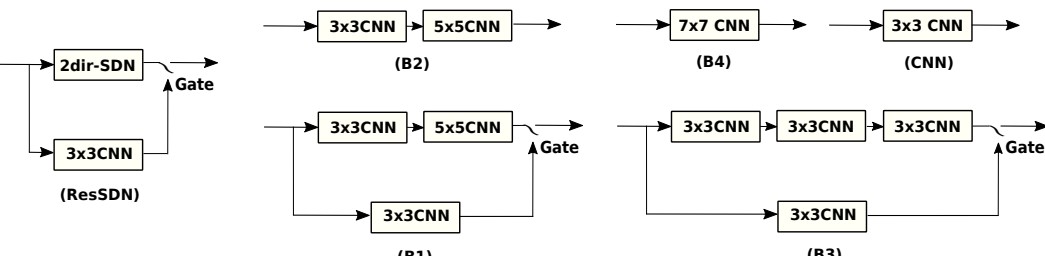

Figure 8: **ResSDN layer and the baseline blocks.**

All baseline blocks were trained in the same setting as SDN-VAE i.e. all the parameters were kept fixed, except from the learning rate, which we halved after every 10 unsuccessful attempts to stabilize the training. IAF-VAE based on the CNN block was stable at the default learning rate of 0.002, B2 was stable at the learning rate of 0.001, while B4 and B1 were stable only at 0.0005. B2 was unstable. We also tested ResSDN at a learning rate of 0.001 to understand if there would be a drop in performance, but there wasn't any except from marginally slower convergence time.

The best runs in terms of negative evidence lower bound (ELBO) for each of the baseline blocks, along with our most successful run for SDN-VAE architecture are reported in Table 4. What we can observe is that the increase in capacity can indeed be correlated with good performance. However, even those VAE models which contained more parameters than the proposed SDN-VAE architecture were not able to converge to a higher negative ELBO. In fact, there exist a considerable performance gap between baseline blocks and the proposed ResSDN layer.

|  | Layer | Number of VAE parameters in millions | Time to converge in hours | in iterations | Best Neg. ELBO in BPD |
|---|---|---|---|---|---|
| **Baselines** | CNN | 42M | 17h | 140K | 3.081 |
|  | B1 | 192M | 21h | 75K | 3.080 |
|  | B2 | 104M |  | unstable |  |
|  | B3 | 126M | 23h | 90K | 2.945 |
|  | B4 | 130M | 22h | 124K | 3.120 |
| **Ours** | ResSDN | 121M | 45h | 60K | 2.906 |

Table 4: **The comparison of ResSDN layer to the baselines from Figure 8.** We replaced CNN layers in the IAF-VAE+ architecture (from Section 4) with baseline blocks and compared the density estimation performance in terms of ELBO. The experiments were conducted on CIFAR-10 dataset.

## A.3 Additional SDN Analysis

**Number of parameters.** For a filter size of 200 (a value used in the SDN-VAE experiments), we compare the number of parameters between CNN and SDN layers. Note that the input scale does not have any effect on the resulting numbers. The numbers are given in Table 5. Here 'dir' denotes the directions in SDN. 'Project phase' denotes the size of project-in and project-out SDN sub-layers which are in this experiment of the same size, since no upsampling is performed. We can observe that the 2-directional SDN is approximately of the same size as 5x5CNN in terms of number of free parameters.

|  | 3x3CNN | 5x5CNN | Project phase | SDN cell | 1dir-SDN | 2dir-SDN |
|---|---|---|---|---|---|---|
| # parameters | 360200 | 1000200 | 40200 | 481200 | 561600 | 1042800 |

Table 5: **Number of parameters of different neural layers.**

**Runtime unit tests.** We measured the execution times of forward propagation of CNN and SDN layers, for different input scales. To obtain standard deviation estimates, each forward propagation was repeated 100 times. The batch size was set to 128. The results are given in Table 6. The SDN layer is considerably slower than the CNN layer, but in the reasonable limits. Note that more efficient implementation will likely improve SDN runtime performance.

| Input scale vs. Layer | 3x3CNN | 5x5CNN | 1dir-SDN | 2dir-SDN |
|---|---|---|---|---|
| 4 | 0.32±0.01 | 1.26±0.01 | 1.93±0.06 | 3.52±0.07 |
| 8 | 0.63±0.01 | 3.25±0.13 | 4.09±0.06 | 7.61±0.09 |
| 16 | 2.33±0.08 | 11.2±0.08 | 11.4±0.03 | 21.0±0.05 |
| 32 | 9.18±0.16 | 20.4±0.18 | 45.4±0.09 | 83.8±0.15 |

Table 6: **Runtime unit tests.** The execution time of forward propagation in ms.

## A.4 Image Synthesis – Additional Results

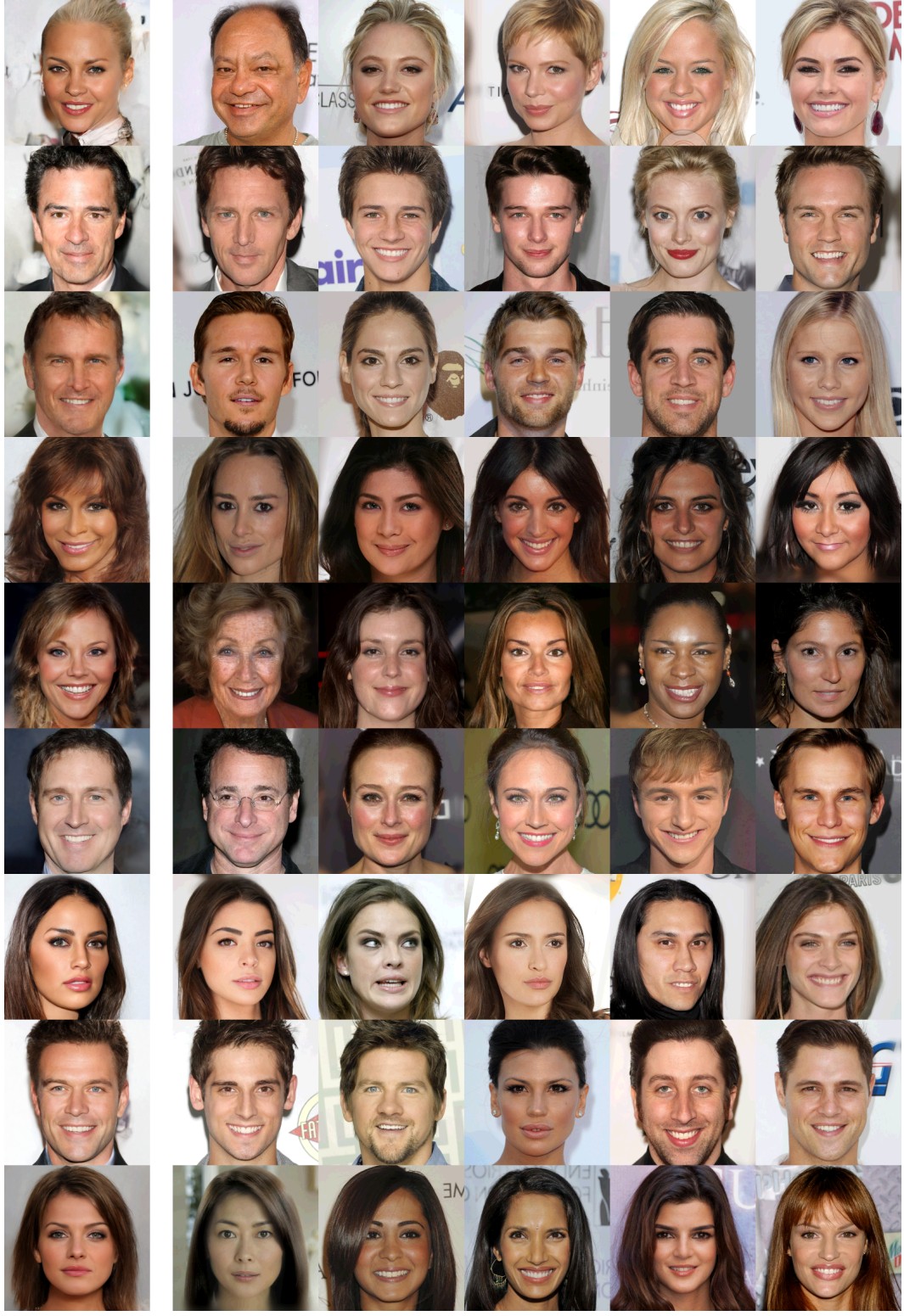

Figure 9: **MSE-based nearest neighbors.** On the left, shown are images from Figure 6. On the right, shown are nearest neighbors in terms of mean squarred error (MSE), found in the training data set. There are no signs that generated samples are replicas of the training ones.

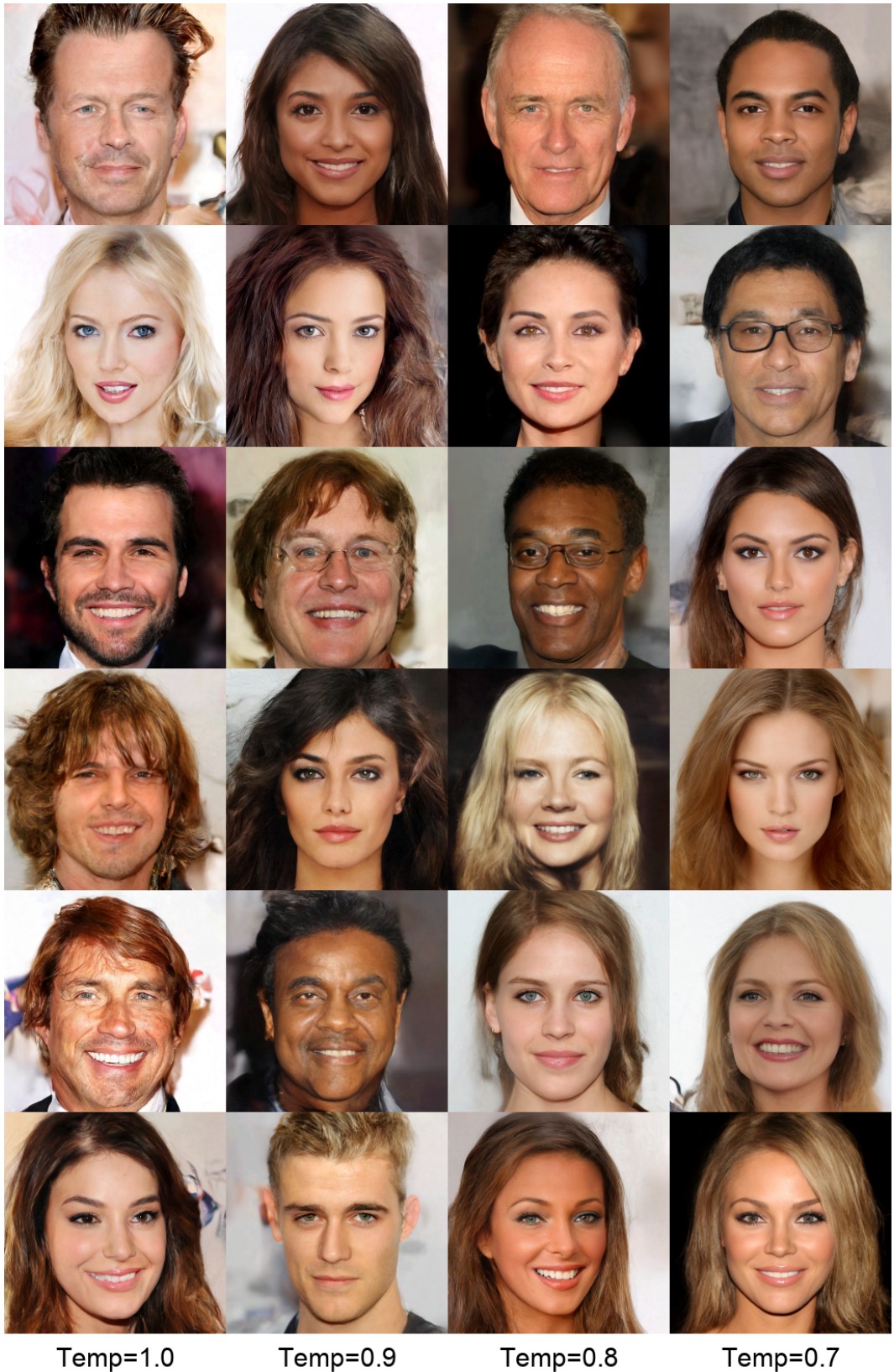

Temp=1.0    Temp=0.9    Temp=0.8    Temp=0.7

Figure 10: **Additional samples synthesized at different temperatures.** As we decrease the temperature, getting closer to the mean of the prior, the photographs become smoother i.e. more 'generic'.

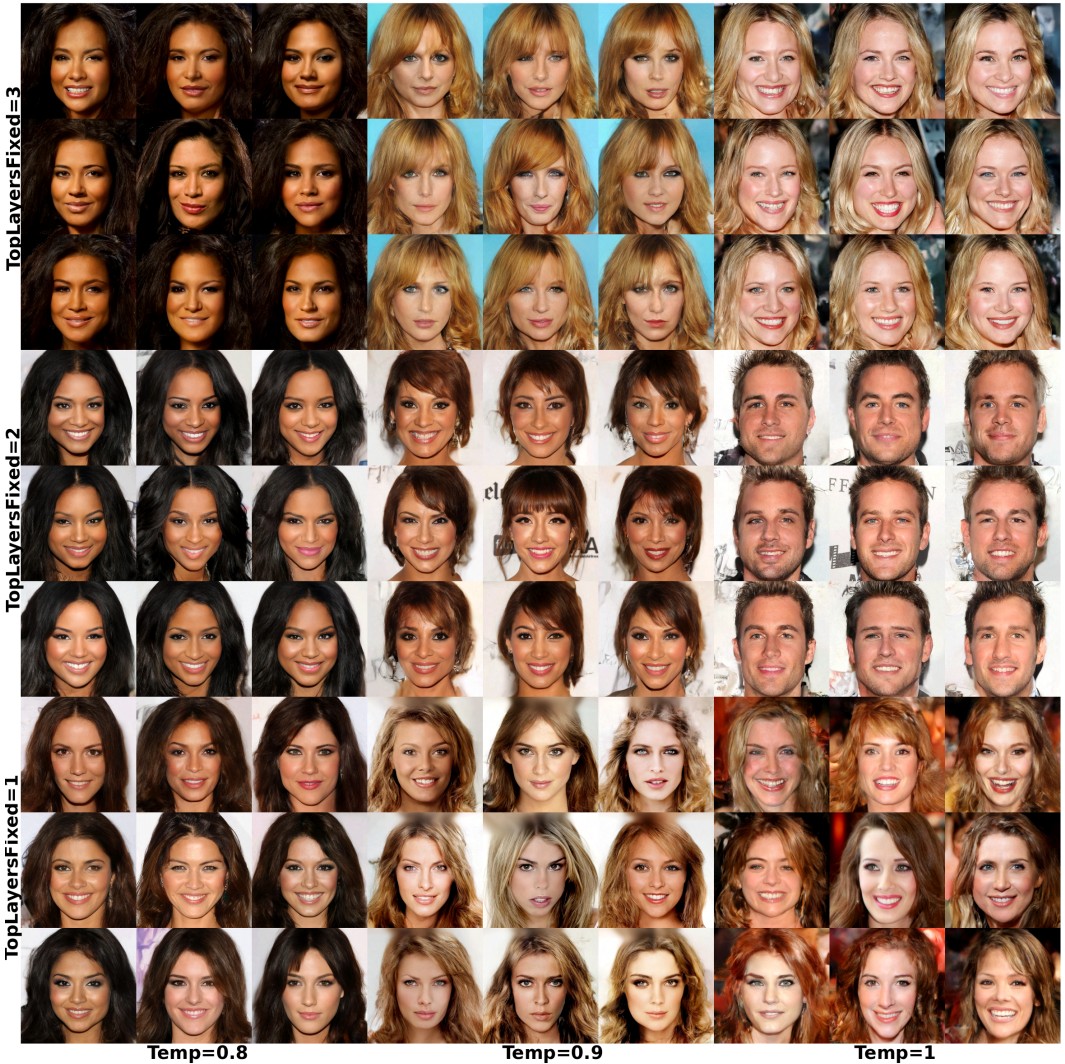

Figure 11: **Sampling around a test image, for varying temperatures and number of fixed layers.** Presented is a 3x3 grid. In each grid there is a 3x3 sub-grid in which an image in the center is encoded, and the surrounding images are sampled conditioned on the specified number of layers from the encoded image and at the specified temperature. The text on the left hand side denotes the number of layers taken from an encoded image to condition sampling on. The text on the bottom denotes the temperature at which samples were drawn. Two key observations can be made: *(a)* lowering the temperature decreases the level of details, making photographs smoother; *(b)* Most of the information in SDN-VAE is encoded in the topmost layer. As we go further down the chain of the generator network, there is a gradual decrease in information content in latent stochastic variables.

### A.5 LEARNING DISENTANGLED REPRESENTATIONS – EXPERIMENT DETAILS

The hyperparameter configuration is given in Table 7. In principle, we followed Higgins et al. (2016); Locatello et al. (2019) in configuring parameters. Notable changes include using of dicretized logistics (Kingma et al., 2016) as the observation model and the Adamax optimizer. We also applied $\beta$-VAE annealing to avoid posterior collapse early in the training, an issue for both considered architectures. The base architecture was again taken from the same previous works. The exact description of architectures is given in Table 8 for the CNN-based VAE, and in Table 9 for the SDN-based VAE.

|  | 3D Shapes $64^2 = 64 \times 64$ |
|---|---|
| # data samples | 448K |
| SDN #channels | 200 |
| Number of directions per SDN | 1 |
| Optimizer | Adamax |
| Learning rate | 0.001 |
| Batch size | 128 |
| Total number of training iterations | 200k |
| Prior and posterior models | Gaussian |
| Observation model | Discretized Logistics |

Table 7: **Configuration of disentangled representation learning experiments.**

| Encoder | Decoder |
|---|---|
| $4 \times 4$ conv, 32 ReLU, stride 2 | FC, 256 ReLU |
| $4 \times 4$ conv, 32 ReLU, stride 2 | FC, $4 \times 4 \times 256$ ReLU |
| $4 \times 4$ conv, 64 ReLU, stride 2 | $4 \times 4$ upconv, 64 ReLU, stride 2 |
| $4 \times 4$ conv, 64 ReLU, stride 2 | $4 \times 4$ upconv, 32 ReLU, stride 2 |
| FC 256 | $4 \times 4$ upconv, 32 ReLU, stride 2 |
| FC $2 \times 10$ | $4 \times 4$ upconv, 3 (number of channels) , stride 2 |

Table 8: **CNN-based vanilla VAE architecture.**

| Encoder | Decoder |
|---|---|
| $4 \times 4$ conv, 32 ReLU, stride 2 | FC, 256 ReLU |
| $4 \times 4$ conv, 32 ReLU, stride 2 | FC, $4 \times 4 \times 256$ ReLU |
| $4 \times 4$ conv, 64 ReLU, stride 2 | $4 \times 4$ upconv, 64 ReLU, stride 2 |
| $4 \times 4$ conv, 64 ReLU, stride 2 | $4 \times 4$ upconv, 32 ReLU, stride 2 |
| FC 256 | 1dir-SDN with 200 channels |
| FC $2 \times 10$ | $4 \times 4$ upconv, 3 (number of channels) , stride 2 |

Table 9: **SDN-based vanilla VAE architecture.**

## A.6 LEARNING DISENTANGLED REPRESENTATIONS – ADDITIONAL RESULTS

For the sake of completeness, we also provide learning curves for varying values of $\beta$ and for different individual terms of $\beta$-VAE objective function. The plots are shown in Figure 12.

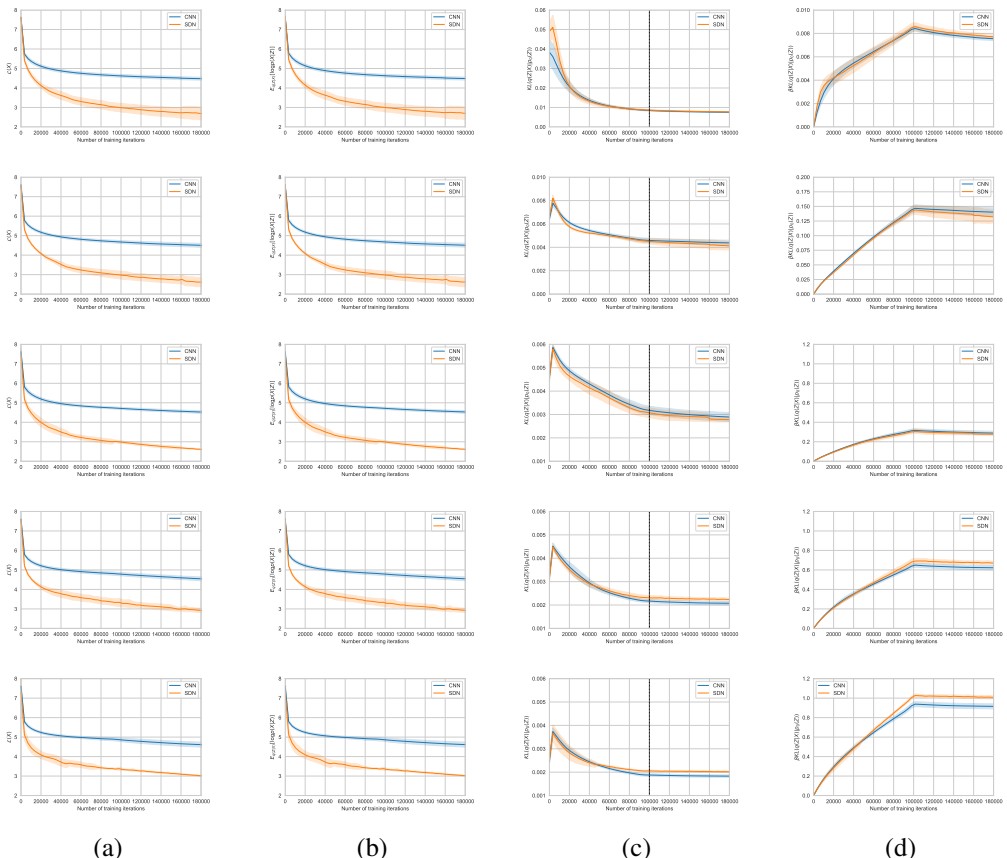

Figure 12: **$\beta$-VAE learning curves for CNN and SDN-based VAE architectures.** Presented are the following quantities measured over the course of training: *(a)* evidence lower bound (ELBO); *(b)* conditional log-likelihood (reconstruction) term; *(c)* KL divergence term; *(d)* $\beta$-scaled KL divergence term; Top-to-bottom, the rows are related to $\beta = \{1, 32, 100, 300, 500\}$. Each pair ($\beta$ value, VAE architecture) is trained for 10 different random seeds. Linear $\beta$-annealing procedure was performed, where $\beta$ was increased from 0 to its final value across the span of 100K training iterations (the end of the annealing procedure is denoted by the vertical line in the plots of the column (c)).

