# OpenReview forum: "Spatial Dependency Networks: Neural Layers for Improved Generative Image Modeling"
_ICLR.cc/2021/Conference — ICLR 2021 Poster_

### Official Review · AnonReviewer3 · 2020-10-19
**A nice idea with computing time drawback**

**Rating:** 7
**Confidence:** 3

**Review:**

1. Summary. This work is in line with the latest attempts[1,2] to improve the VAE quality by employing more powerful decoding architectures. The authors propose a new block that handles (non-local) spatial dependence of pixels at the expense of increased computation time - O(scale). The presented experiments demonstrate that usage of the suggested layer is beneficial for the tasks of density estimation and image generation.
1. Decision. The described idea is novel, clear, and supported by experiment results. To my mind, the provided evaluation is thorough enough, the ablation study is quite comprehensive. Therefore, I tend to vote for acceptance. However, major issues of this approach are the increased number of parameters and computational time. This can make the applicability to other datasets limited.
1. Questions.
    1. Is it sufficient that the sweep from Algorithm 1 is GRU-like, or, more generally, gated? Could it be replaced with a masked-convolutional sweep with the output depending only on the "past" (row number <= i) and the pixel (i, j)'s value itself?
    1. Am I right you employed Monte-Carlo integration to provide NLL in Table 1 for VAE-based models? How many samples did you use?
1. Remarks. I believe the headings of Tables 2 & 4 are a bit misleading: ELBO is a lower bound (the higher score - the better), however in your notation the less - the better.

[1] https://arxiv.org/abs/2007.03898
[2] https://openreview.net/forum?id=RLRXCV6DbEJ

#############

After reading the rebuttal I confirm the initial rating.

---

> ### Author Response · Authors · 2020-11-14
> **SDNs could be: (i) integrated into any new VAE architecture; (ii) implemented in a different way;**
>
> We thank the reviewer for the feedback, and for praising the novelty of the idea, the write-up, and the experimental evaluation. We would like to reply to the specific remarks.
>
> **"major issues of this approach are the increased number of parameters and computational time. This can make the applicability to other datasets limited"**
>
> The computational time during training is indeed the central roadblock at the moment, and in our draft, we try to be honest about it. We analyze this in more detail in the main post given above, in the 'Runtime' section. The number of parameters is something to be less concerned about, as explained in the 'Model size' section of the main post. We will update the draft accordingly.
>
> **"Is it sufficient that the sweep from Algorithm 1 is GRU-like, or, more generally, gated? Could it be replaced with a masked-convolutional sweep with the output depending only on the "past" (row number <= i) and the pixel (i, j)'s value itself?"**
>
> -We empirically found gated to be superior to non-gated architectures. We also tested alternative architectures, LSTM, and vanilla RNN. However, the one based on GRU offered the best performance-memory trade-off. Naturally, a simpler gating unit has fewer parameters hence reduces the memory of the modeled VAE (in that respect GRUs, are preferable to LSTMs).
>
> -Indeed, a possible implementation of SDN would be to use masked convolutions. However, unlike e.g. PixelCNNs which leverage on teacher forcing due to autoregressive modeling, we cannot circumvent sequential computation, so the CNNs with different masks would have to be executed sequentially. One could, for example, stack masked CNNs on top of each other, but that would be memory consuming as we would need as many masked CNNs as there are number of sweeps. In SDN instead, we have a single SDN cell instead of a stack of CNNs, which is more memory efficient.
>
> **"Am I right you employed Monte-Carlo integration to provide NLL in Table 1 for VAE-based models? How many samples did you use?"**
>
> As explained in Table 3 in the row 'Number of importance samples', the number of MC samples was 1024.
>
> **"I believe the headings of Tables 2 and 4 are a bit misleading: ELBO is a lower bound (the higher score - the better), however in your notation the less - the better."**
>
> This is correct. In the updated draft it will be changed to 'Negative ELBO'. Thank you for pointing this out.
>
> **"This work is in line with the latest attempts[1,2] to improve the VAE quality by employing more powerful decoding architectures."**
>
> This is also correct. The referenced works are complementary to ours. For example, one could use [1] or [2] as a backbone architecture instead of IAF-VAE, replacing CNNs with SDNs. Our work is also more general as SDNs can be applied to e.g. GANs or image-to-image translation.

---

### Official Review · AnonReviewer4 · 2020-10-25
**Good results; discussion/comparison to related work could be improved**

**Rating:** 6
**Confidence:** 3

**Review:**

Summary: The authors propose a new building block called Spatial Dependency Networks (SDN) that increases the expressiveness of a CNN at the cost of some sequential computation. SDN propagates information across a 2D feature map using LSTM-like updates across the rows (or columns). They use SDN blocks to improve the decoder architecture of IAF-VAE [1], a SOTA VAE-based method, and show a substantial performance boost on generative model tasks. Their method performs much better than other VAE methods, and almost as well as SOTA autoregressive models.

Strengths:

Solid empirical results & thorough experiments
Paper is mostly well-written & many experimental details are included.
Weaknesses:

The motivation seems a bit fuzzy to me: the primary contribution of this work seems to be the SDN building block, but it seems like SDN isn't necessarily limited to generative modeling? The authors clearly discuss the advantages of SDN over CNN, but the experiments focus on generative modeling VAEs, but not any of the other tasks where CNNs are used & more expressive architectures might be helpful.
The discussion of related work is quite limited, in particular with respect to autoregressive models. The SDN block is quite reminiscent of the Row LSTM from PixelRNN [2], a pioneering autoregressive model. Autoregressive models have been known to perform the best density estimation (compared to other families of likelihood-based generative models), at the expense of slow sampling speed (since each element must be sampled sequentially). Since SDN introduces some sequential dependencies, it trades off computation during sampling time vs during likelihood evaluation (autoregressive models are slow at the former but fast at the latter; the proposed method is somewhere in the middle but the same speed for both). I would like to see some more discussion along these directions to better place the authors' method into context with related work.
On a similar note, the authors allude to the extra computational overhead incurred by adding SDN to VAEs, but I don't think they explicitly state what that is? This would also help frame the contributions of their work.
Nit: the description of SDN in Algorithm 1 is a bit confusing -- it doesn't seem to convey the sequential nature of SDN? From reading Algorithm 1 only, it sounds like SDN could be implemented via masked convolutions + gating, but this seems to disagree with the rest of the paper? Also, the output variable I_+ doesn't seem to be used?
Overall, I would slightly lean towards acceptance (conditioned on improving the discussion around the points from above), but am curious to see what the other reviewers think.

---

> ### Author Response · Authors · 2020-11-14
> **We will revise motivation, related work and computational requirements**
>
> We thank the reviewer for the feedback, and for praising the quality of the write-up and the experimental design and results. Below we discuss the specific remarks.
>
> **"The motivation seems a bit fuzzy to me: the primary contribution of this work seems to be the SDN building block, but it seems like SDN isn't necessarily limited to generative modeling? The authors clearly discuss the advantages of SDN over CNN, but the experiments focus on generative modeling VAEs, but not any of the other tasks where CNNs are used and more expressive architectures might be helpful."**
>
> We now realize that the motivation and the potential use cases of SDN should be clarified further. We dedicated the section "Motivation, applications and scope" in the main post.
>
> **"The discussion of related work is quite limited...The SDN block is quite reminiscent of the Row LSTM from PixelRNN [2]...Autoregressive models have been known to perform the best density estimation (compared to other families of likelihood-based generative models), at the expense of slow sampling speed (since each element must be sampled sequentially). Since SDN introduces some sequential dependencies, it trades off computation during sampling time vs during likelihood evaluation (autoregressive models are slow at the former but fast at the latter; the proposed method is somewhere in the middle but the same speed for both). "**
>
> To this, we dedicated the 'Related work' section in the main post. SDNs exploit the inductive bias of spatial dependencies, inspired by how autoregressive models work. However, because the sequential process is done in the latent space, the generation order of pixels is not preordained -- pixels are generated "at once", not one-by-one. The reviewer is correct in observing that some autoregressive architectures are faster during training: this is because they leverage on teacher forcing -- however, the unfortunate consequence of this are the phenomenas such as 'posterior collapse'. This does not happen in SDNs as mentioned in our disentanglement experiments. Finally, in the sampling time, the SDN-VAE combination is faster than autoregressive architectures, as correctly noted by the reviewer. We suggest inserting this discussion in the updated draft. We also refer the reviewer to our discussion with reviewer 1.
>
> **"On a similar note, the authors allude to the extra computational overhead incurred by adding SDN to VAEs, but I don't think they explicitly state what that is?"**
>
> We refer the reviewer to the 'Runtime' and 'Model size' sections of the main post, and also to Table 4 in the original draft.
>
> **"The description of SDN in Algorithm 1 is a bit confusing -- it doesn't seem to convey the sequential nature of SDN? From reading Algorithm 1 only, it sounds like SDN could be implemented via masked convolutions + gating, but this seems to disagree with the rest of the paper? Also, the output variable I_+ doesn't seem to be used?"**
>
> In Algorithm 1, there is a loop i=1:N in which SDN sweeps over the feature map 'correcting' feature vector estimates one by one, hence the complexity $\mathcal{O}(N)$. In that sense, SDN is done sequentially, which is in coherence with the rest of the paper. For the discussion on masked convolutions, we refer the reader to our response to reviewer 3. The potential source of the confusion might be the loop j=1:N which only shows that columns/rows are updated in parallel.
> Finally, $I_+$ is used in the 'project-out' phase and simply represents the a posteriori value of the intermediate feature map I, after the correction phase.

---

### Official Review · AnonReviewer2 · 2020-10-27
**Interesting approach that seems to improve performance in VAEs (at the cost of longer training/inference time)**

**Rating:** 7
**Confidence:** 3

**Review:**

This paper introduces a new neural network layer architecture (spatial dependency network - SDN) that can be used in place or in conjunction with traditional convolutional layers. The SDN layers' main difference to convolutional layers is that in SDN layers extracted features are dependent and influenced by nearby features. SDNs are evaluated with VAEs on density modeling and learning of disentangled representation and show better performance than purely convolutional baseline architectures.

Strengths:
- The new layer architecture can be used as a drop-in replacement for convolutional layers and shows promising results on the evaluted tasks
- Evaluation is done on two different tasks and shows that SDNs perform better on both tasks than convolutional VAEs
- Using SDNs seems to support sampling from VAEs at much higher temperatures than usually possible

Questions/remarks:
- What is the relationship (similarities/differences) between SDN and normal (self-)attention? Why would SDN be better than attention (except maybe faster runtime)? What can SDN do that attention can't (in principal or in practice)?
- Section 2, "project-in" stage: why are the project-in and project-out states needed? Can't the corrections stage be performed on the input features directly?
- Figure 2 (b): "Solid arrows represent direct dependencies, and dashed indirect ones" -> what exactly do you mean with "indirect dependencies"?
- Avoiding vanishing gradients: do you actually ever observe vanishing gradients? have you tried training with/without any normalization methods to see if vanishing gradients become a problem?
- SDN-VAE architecture: why do you only use SDN in the decoder? Have you tried using them also in the encoder?
- How much overhead does SDN in the SDN-VAE actually introduce in practice - regarding model size/VRAM and training time? How much bigger is the SDN-VAE compared to the baseline and how much longer does it take to train/per iteration? I see some of these answers are given in the tables in the appendix but I think it would be beneficial to be open about this and mention the trade-off in the main part of the paper.
-Learning disentangled representations: why do you think SDN helps with learning disentangled representations?

Overall I think this is a good paper. It would be interesting to see further results with SDN, e.g. on a baseline DCGAN architecture, to see if the improvements transfer to other architectures.
I also believe the paper would benefit from a discussion comparing SDNs to traditional attention and when SDNs might be a better choice than attention. It would also be interesting to know a comparison of attention  vs SDN in terms of training and run-time. Attention has a higher complexity in O notation but it is not clear what exactly this means in practice, especially when attention/SDN are only used on "small" feature layers early on in the decoder.

###
###

Update after revision: I have looked over the revised paper and believe the authors have addressed most issues that were raised by the reviewers, especially by describing in more depth the relation of their approach with autoregressive models and self-attention and mentioning the runtime differences of their model compared to a normal CNN.
I have, therefore, raised my score to 7.

---

> ### Author Response · Authors · 2020-11-14
> **We will add a discussion on relation to self-attention and computational requirements**
>
> We thank the reviewer for the feedback, and for emphasizing the possible impact of SDN and the quality of results.
>
> **"What is the relationship (similarities/differences) between SDN and normal (self-)attention? Why would SDN be better than attention (except maybe faster runtime)? What can SDN do that attention can't (in principal or in practice)?"**
>
> We will add a dedicated explanation in the updated draft. The similarity of attention mechanism and SDN is the ability to create long-range dependencies within the feature maps. The key difference is how these dependencies are modeled: in SDNs only neighboring feature vectors are dependent on each other in the computational graph, and the dependencies between distant feature vectors are 'indirect' in the sense that there is a chain of direct dependencies between them. SDNs use gated units to ensure that the information is propagated across large distances. On the other hand, self-attention requires an additional mechanism to incorporate positional information, which makes it more difficult to enforce spatial coherence but 'easier' to capture long-range dependencies. Runtime-wise, the complexity is $O(N^2)$ instead of $O(N)$ as correctly noted by the reviewer. In our experiments, we applied multiple layers of SDNs, even at a scale 64 so we hypothesize that the training time would significantly deteriorate with self-attention. Finally, these models could be seen as complementary: self-attention is better in capturing long-range spatial dependencies while SDN is better in enforcing spatial coherence.
>
> **"Why are the project-in and project-out states needed? Can't the corrections stage be performed on the input features directly?"**
>
> To some extent, these stages exist for practical convenience. The project-out stage converts feature vectors from (-1,1) range, as constrained by 'tanh', to (-inf,+inf). This allows one to use SDN as a drop-in replacement for CNN. The project-in stage is used to perform upsampling when necessary. In both stages, the number of feature vectors can be increased/reduced -- this helps controlling overfitting and memory consumption e.g. we used different SDN sizes at different scales in our experiments. Also, in practice, one can stack an arbitrary number of correction stages between the two project stages. We do not exclude the possibility that there exists a better variation of the proposed architecture which might be exploited in the future.
>
> **"Figure 2 (b): "Solid arrows represent direct dependencies, and dashed indirect ones"**
>
> By indirect dependencies, we refer to the dependencies between two feature vectors which are not neighboring, but the dependency exists through a sequence of intermediate feature vectors (see Fig2b and also our response to the first remark). We will add additional text to clarify this.
>
> **"Avoiding vanishing gradients: do you actually ever observe vanishing gradients? have you tried training with/without any normalization methods to see if vanishing gradients become a problem?"**
>
> In our SDN-VAE experiments, we found that gradients were at times, at the beginning of the training, not propagating to deeper layers, independently whether we used weight normalization or not. This problem was remedied by inserting residual (highway) connections. In essence, residual connections indeed improved training stability. In our disentanglement experiments, we did not encounter this problem.
>
> **"SDN-VAE architecture: why do you only use SDN in the decoder? Have you tried using them also in the encoder?"**
>
> This is somewhat related to the 'Motivation, applications and scope' section in our main post. The intended use case of SDNs are deep nets which produce images, as the output of SDNs should exhibit the property of spatial coherence. If we take disentanglement experiments for example, there is no obvious reason why latent variables of the encoder should be spatially coherent, especially given that in the end we aim to achieve statistical independence between latent dimensions. While this was out of our experimental scope, in the future it could be explored in more depth.
>
> **"How much overhead does SDN in the SDN-VAE actually introduce in practice..?"**
>
> We refer the reviewer to the 'Runtime' and 'Model size' sections of the main post.
>
> **"Learning disentangled representations: why do you think SDN helps with learning disentangled representations?"**
>
> SDNs can be thought of as a tool that introduces an inductive bias in order to improve the convergence of the optimizer. If the optimization objective is the modified $\beta$-VAE ELBO, then the final $\beta$-VAE modified ELBO after optimization will be improved, meaning that we are likely to obtain better disentanglement. One can also think about the answer in a different way: a more expressive SDN decoder 'relieves' the pressure from the encoder which can then focus on decorrelating latent dimensions rather than just transmitting the information through the bottleneck.

---

> > ### Comment · AnonReviewer2 · 2020-11-20
> > **Thank you.**
> >
> > Thank you for the detailed comments.
> > I look forward to the updated manuscript. I think most of the points (by other reviewers and me) were addressed by your answers and incorporating/highlighting this in the manuscript will improve it even more.

---

### Official Review · AnonReviewer1 · 2020-10-27
**Encouraging results, but perhaps somewhat limited in scope**

**Rating:** 6
**Confidence:** 5

**Review:**

**Summary**: This paper proposes a network architecture, spatial dependency network (SDN), that attempts to more explicitly model spatial dependencies, as compared with convolutional networks. The network involves three steps: projection (via 1x1 CNN), a spatially (by row/column) autoregressive transform applied in each direction, and another projection. The authors use the architecture in place of CNN layers in the IAF VAE architecture. However, note that this is in the conditional mappings and *not* the flows. The model achieves state-of-the-art performance, as compared with non-autoregressive models. The authors also demonstrate improved disentanglement on a single-level VAE.

**Strong Points**: The technical aspects of the approach are generally well-described and clear. I found the diagrams and algorithm box helpful in understanding SDNs. Likewise, the adaptation of the diagram from Kingma et al., 2016 was helpful to distinguish which components of the architecture were modified in SDN-VAE over the IAF architecture. The basics of variational inference and VAEs are described concisely and correctly.

While I did not find the experimental results entirely unsurprising (see below), the experiments do appear to be rigorous. The authors demonstrate their network within a relevant architecture (IAF) on multiple benchmark image datasets. Results are compared with relevant recent approaches, where SDN-VAE outperforms non-autoregressive models. While this might be attributed to an increased number of parameters, the authors also conduct an ablation study to demonstrate that SDNs outperform much larger CNN architectures. This suggests that SDNs truly provide a useful inductive bias toward modeling dependencies between dimensions, analogous to how ResNet architectures typically outperform deeper non-ResNet architectures.

The results on latent factorization are also rigorous. The authors evaluate performance with multiple $\beta$ values and report latent factorization using two metrics. Results are reported using 10 random seeds, helping to confirm that the improvement in disentanglement is significant.

The results generally seem to be reproducible, as algorithmic and architectural details are present in the main paper and the appendix. The main results (SDN-VAE) are demonstrated using an existing model architecture (IAF), so the majority of other modeling design choices are already publicly available.

The paper is formatted well, with clear titles for sections, figures, tables, etc. This will be helpful for unfamiliar readers.

The approach is not entirely novel (see below), but admittedly, many current VAEs exclusively use convolutions in the generative and inference models. This paper, while also using convolutions, introduces autoregressive transforms throughout the conditional mappings. This is potentially a worthwhile contribution, moving the generative modeling community toward other forms of functions (beyond feedforward and convolutional) in the mappings.

**Weak Points**: While I agree with the approach, I felt as though it was poorly motivated. The authors attempt to motivate SDN by discussing the importance of modeling spatial dependencies between pixels in images, often appealing to notions of “coherence.”  For instance, the authors claim that SDNs “explicitly model image-level spatial coherence and dependencies.” However, from what I can tell, coherence is never defined. I’m assuming that the authors are referring to modeling the full joint distribution over dimensions, rather than assuming independence between dimensions. (Note that VAEs can already model such dependencies through their latent variables.) The closest that the authors come to explicitly motivating the difference from previous approaches is in the “Comparison to CNNs” in Section 2. However, this discussion incorrectly casts the network input/output as random variables, when they are, in fact, parameterized by deterministic functions.

The approach is somewhat lacking in novelty. Admittedly, the particular architecture itself is novel, and it is likely a useful improvement. However, it belongs to the larger class of neural autoregressive density estimators (NADE) and autoregressive models. Effectively, this is replacing a direct mapping with an autoregressive mapping, which has been used before, e.g. PixelCNN decoders in VAEs. The authors attempt to distance their work from autoregressive models in Section 2, saying that these are “conceptually very different,” yet autoregressive models can also contain multiple orderings of dependencies. This is the idea behind permutation/reversal operations found in autoregressive normalizing flows (which generalize Gaussian autoregressive models).

Given the similarity with previous autoregressive modeling approaches, I found it surprising that the authors chose to parameterize the conditional mappings, rather than the distributions, with SDNs. This is particularly surprising given that the authors chose the IAF architecture as their baseline, which includes autoregressive normalizing flows at each level of latent variables. While it’s unclear which form of model should be preferred, it seems like an obvious choice to at least evaluate using SDNs to parameterize the normalizing flows (e.g. IAF used MADE to parameterize the autoregressive flows).

More generally, I found it surprising that the authors chose one particular model class to demonstrate their network architecture. Convolutional networks are used widely in many classes of generative models, as well as discriminative models. If SDNs are a swap-in replacement for CNNs that are truly better at capturing spatial dependencies, then this should be apparent across multiple settings, not just latent variable models (VAEs). Experiments across multiple tasks and model families would help to improve the impact of the paper.

The main experimental results of the paper are 1) improved generative modeling on image datasets, and 2) improved factorization in single-level models. Given that SDNs are more powerful than CNNs, neither of these results are unexpected. Assuming one can avoid local optima, an improved conditional mapping should yield better density estimates, as well as map a standard Gaussian to data estimates (factorization). I do not expect every experimental result in a paper to be surprising, but it’s unclear what we learn from the results in this paper (other than the fact that SDNs work in practice).


**Accept / Reject**: I found this paper somewhat difficult to assess. While the SDN-VAE model achieves state-of-the-art log-likelihood results on multiple image datasets, the method itself is somewhat lacking in novelty and does not seem to appreciably improve our understanding of these models. Autoregressive transforms are already heavily utilized in generative modeling, so it is generally unsurprising that including more autoregressive computations in the model will improve performance. Similarly, it seems fairly intuitively obvious that an improved conditional generative mapping should improve disentanglement, as it can more easily warp a (disentangled) Gaussian random variable into the data distribution.

Further, the authors present SDNs as a swap-in replacement for CNNs, however, experiments are only performed in one setting: explicit latent variable models of images. Spatial dependencies are relevant for any image dataset, and other data modalities more broadly. This paper would benefit from exploring these other settings (other generative models and discriminative models). Similarly, I would have liked to see the authors use their autoregressive transform within normalizing flows. This seems like a natural use-case.

The above points aside, developing improved network architectures is still a worthwhile endeavor. I’m hopeful that SDNs could become a standard network layer within the image modeling community. More broadly, it may help researchers to consider spatial dependencies more explicitly when constructing network architectures. For these reasons, I am slightly in favor of accepting this paper.

**Additional Feedback**:

Abstract:
The first sentence is grammatically incorrect. It is missing a subject.

Introduction:
It would help to clarify what is meant by the word “coherent.”
Similarly, it would help to briefly describe why CNNs (supposedly) cannot explicitly model spatial dependencies and coherence.
Figure 1: should label the convolutional filters. Also, I understand that this is a generic overview of SDN-VAE, but it is somewhat too simplistic to be useful.

SDN Architecture:
I found some of the notation confusing. For example, if $\mathbf{B}^{(1)}$ is a matrix, then why is being added to a vector in Eq. 1?
The tanh on the input (in Algorithm 1) could be explained further. Is this for stability?
It would be helpful to discuss an order-of-magnitude comparison of the number of parameters between SDNs and CNNs.
The gated residual is referred to as a “highway” connection.
It seems inaccurate to express the output of SDNs as a probability (though I understand the point).
In related work, I would discuss (autoregressive affine) normalizing flows, e.g. IAF, MAF, etc., and autoregressive neural networks, e.g. NADE, MADE, EoNADE, etc. These seem highly related.

SDN-VAE:
Should also cite Rezende et al., 2014 for the reparameterization trick.
Should mention that the approximate posterior in top-down inference is structured, i.e. it models dependencies between levels of latent variables.

Image Density Modeling:
I would not necessarily consider density estimation and image synthesis as different “tasks.”
In the maximum likelihood objective, missing a $1 / N$ factor.

Learning Disentangled Representations:
I would disagree that non-hierarchical models are more suitable for representation learning. Hierarchical models generally extract increasingly abstract features, which can be quite useful.

Conclusions:
Somewhat misleading to compare SDN-VAE with GANs. There is nothing intrinsic to SDNs that allows them to calculate explicit density estimates. Indeed, SDNs could also be used in GANs.

---

> ### Author Response · Authors · 2020-11-14
> **We will update the motivation, related work and computational requirements**
>
> We thank the reviewer for the extensive feedback and clarity of critics, and also for highlighting the quality of the writeup and experiments.
>
> ##### Weak points
>
> **"I felt as though it was poorly motivated...from what I can tell, coherence is never defined"**
>
> Since this is a shared concern, we refer the reviewer to the section "Motivation, applications and scope" in the main post.
>
> **"..this discussion incorrectly casts the network input/output as random variables, when they are, in fact, parameterized by deterministic functions."**
>
> We agree that $p(X^{L_{k+1}}|X^{L_k})$$= \prod_{i,j} p(X^{L_{k+1}}(i,j) | X^{L_k})$ may be misleading as $X^{L_{k+1}}|X^{L_k}$ is a deterministic function. However, casting input/output as random variables $X^{Lk}$/$X^{L_{k+1}}$ is legitimate: a deterministic function of a random variable (in this case a random input image or a random latent code) is also a random variable. An alternative notation would be using the language of conditional independence $A \perp B | C $ or mutual information $I(A;B|C)$.
>
> **"The approach is somewhat lacking in novelty..it belongs to the larger class of neural autoregressive density estimators (NADE) and autoregressive models..autoregressive models can also contain multiple orderings of dependencies....it seems like an obvious choice to at least evaluate using SDNs to parameterize the normalizing flows"**
>
> We propose to add the 'Related work' section to the draft and include this comparison. SDNs are indeed inspired by the way AR models exploit spatial coherence (estimating neighboring pixels one-by-one) and model spatial dependencies (exploiting statistical correlations across space). However, in AR models, the generation order of pixels is predetermined due to $\prod p(x_i|x_{<i})$ factorization; one can choose between different orderings, but needs to commit to one at the beginning of the training. In SDN, the sweeps are performed in the latent space, not in the pixel space, and for a single training sample, one can iterate in multiple directions hence capturing the entire grid of dependencies. Furthermore, as we discuss with reviewer 4, there are computational trade-offs. As the reviewer observes, SDNs are indeed 'conditional mappings' and in the context of VAEs, they parametrize the pdf of a decoder. All in all, we believe there is a considerable conceptual gap between NADE and SDNs. Next, normalizing flows impose constraints on invertibility and typically require the diagonal Jacobian, The current form of SDN is not compatible with these. Finally, IAF in IAF-VAE is used to make the posterior more expressive, however, SDN-VAE performs marginally poorer when used without IAF. Similar observation was made in the NVAE paper.
> That said, SDN can be seen as a *novel neural layer* which models spatial dependencies and coherence inspired by how AR *density models* work.
>
> **"Experiments across multiple tasks and model families would help to improve the impact of the paper."**
>
> We fully agree. However, while the application scope of SDN itself is rather large, the scope of this paper is limited to the applications to VAEs as discussed in "Motivation, applications and scope" in the main post.
>
> **"Given that SDNs are more powerful than CNNs, neither of these results are unexpected...it’s unclear what we learn from the results in this paper.."**
>
> One of the goals of this paper was to show that a viable way to make progress in image modeling is to design new architectures, going beyond CNNs. From a practical perspective, our mission was to provide empirical evidence that SDNs outperform CNNs which have been industry standard for many decades. We would like to remind the reviewer that this was not known a priori. Whether SDN will eventually become an industry standard, time will tell.
>
> ##### Technical
>
> **Abstract and introduction**
>
> The technical errors will be fixed, figures too. For 'coherence' see the main post.
>
> **SDN architecture**
>
> Our apologies, it should be vector $\textbf{b}^{(1)}$, also $\textbf{b}^{(3)}$. We will explain tanh in the new draft: the idea is to constrain a priori values of feature vectors to (-1,1) before running the tanh-based gating procedure. 'Model size' is discussed additionally in the main post, as well as connections to AR models in 'Related work'.
>
> **SDN VAE**
>
> We will add the missing citation and clarify the top-down inference details.
>
> **Image Density Modeling**
>
> We will add the missing factor. Image synthesis is indeed a way to explore learned density function. We will update this too.
>
> **Learning Disentangled Representations**
>
> We will clarify that hierarchical models are suitable for this in a different way: factors of variations could be disentangled 'vertically' and not in latent vector dimensions as done in e.g. $\beta$-VAE.
>
> **Conclusions**
>
> We agree that this is misleading. We simply wanted to remind the readers why one would prefer VAEs over GANs in general. This will be corrected.

---

### Author Response · Authors · 2020-11-14
**Main post: Response to all reviewers**

Dear reviewers, thank you for your feedback. We took the liberty to summarize your shared concerns and if you agree, we suggest updating the draft according to the comments below. Other points are discussed in the review-specific threads.

#### **Strong points**

The reviewers are in agreement that our paper is well-written, that experiments are thoroughly designed, rigorously conducted, and reproducible, and that SDNs demonstrate performance improvements in both considered VAE settings. The main contribution is practical: SDNs may be adopted and improved by the community, replacing CNNs in VAE settings and possibly beyond. Additionally, reviewer 1 observes that this research may inspire the community to explore architectures other than CNNs and fully-connected ones.

#### **Points for improvement**

###### **Motivation, applications and scope -- More formal treatment of ’spatial coherence’ is necessary. The lack of clarity makes it difficult to understand in which scenarios SDN is applicable. This makes the motivation of the paper ’fuzzy’**

-We suggest updating the introduction. We would clearly state that SDNs are building blocks for constructing deep nets which at the end of their deep pipeline produce images. If a deep net outputs an image, then the hypothesis is that all the intermediate representations (feature maps) of that image, as well as the image itself, are 'spatial coherent'. If a feature map is thought of as a random variable, then the 'spatial coherence' would imply that the statistical dependencies among feature vectors within that feature map are given in the form of a dependency grid, similar to the one shown in Fig 1, where direct dependencies exist only between neighboring feature vectors. This is somewhat similar to how Markov Random Fields are traditionally used in image segmentation. In SDNs, the inductive bias of spatial coherence is captured through algorithmic design; see the computational flow from Fig 2.

-To that end, SDNs are applicable in the tasks in which images are produced, such as text-to-image and image-to-image translation, image inpainting, image segmentation, denoising, generative modeling, etc. While the application scope of SDNs is broad, this paper exclusively focuses on VAEs. We have tried to be honest about this since the beginning of the paper. Representation learning and density estimation in VAEs were also to our standards computationally demanding; thus we leave other possible extensions and applications for future work.

###### **Related work -- More explicit comparison to related architectures is required. Understanding of the use cases would help assess the novelty of the idea.**

We compare SDN to related work in more detail in our replies below. We suggest adding a 'Related work' section to make a clear distinction to other generative approaches, especially to autoregressive ones: the main difference is that, in our paper, SDN does not model a normalized conditional pdf of an individual pixel. Spatial dependencies are instead captured in latent space, in the decoder of a VAE -- this is not autoregressive image modeling. SDN is simply an unnormalized, nonlinear transformation that can parametrize a pdf in a spatially coherent way.

###### **Model size**

For a filter size 200, we compare the number of parameters for CNN/SDN (note that the input scale does not have any effect):

3x3CNN:  360200

5x5CNN:  1000200

project-in:  40200

SDN cell:  481200

project-out:  40200

1dSDN:  561600

2dSDN:  1042800

The 2-directional SDN is approximately of the same size as 5x5CNN. Also, Table 4 shows that SDN with fewer parameters than a stack of 3x3CNNs achieves better performance.

###### **Runtime**

On a single RTX-GPU, we compare forward propagation runtimes (in ms) for different input scales and for 100 repetitions. The batch size is 128.

3x3CNN: 0.32+-0.01, 0.63+-0.01, 2.33+-0.08, 9.18+-0.16  (for scales 4,8,16,32)

5x5CNN: 1.26+-0.01, 3.25+-0.13, 11.2+-0.08, 20.4+-0.18

1dSDN:   1.93+-0.06, 4.09+-0.06, 11.4+-0.03, 45.4+-0.09

2dSDN:   3.52+-0.07, 7.61+-0.09, 21.0+-0.05, 83.8+-0.15

SDN is slower for 3 reasons: (i) iterations; (ii) gating mechanism; and (iii) not fully optimized implementation; Nevertheless, producing SoTA results in two VAE settings shows that SDNs are usable in the current state. The training times given in Table 3 are comparable to NVAE(https://arxiv.org/abs/2007.03898). Another important observation from Table 4 is that SDN-VAE converged in fewer iterations than CNN-VAEs. Lastly, note that sampling time SDN-VAE is still significantly faster than in AR models, as SDN can operate on smaller scales and is $\mathcal{O}(N)$, while standard AR models operate in pixel space (for HD images, very large scale) and are $\mathcal{O}(N^2)$.

##### **Final remarks**

We would greatly appreciate it if the reviewers responded to the above-suggested modifications. If the reviewers agree, would they be willing to upgrade the scores accordingly?

---

### Author Response · Authors · 2020-11-24
**Updated Manuscript**

Dear reviewers, thank you for helping us improve our manuscript. We included and highlighted the following changes based on our discussion below:

1. Improvements in the introduction and conclusion, to better motivate and explain the idea and the scope of SDN.
2. We added experiments on computational characteristics and discussed them in the main paper to improve transparency.
3. We added the 'Related Work' section to explain relationships with convolutions, autoregressive models, flows, and self-attention.
4. (not all are highlighted) We made some text reformulations and included the replies to your minor remarks.

---

### Decision · Program_Chairs · 2021-01-07
**Final Decision**

**Decision:**

Accept (Poster)

**Comment:**

This work proposes a novel network structure, spatial dependency networks that is introduced as an alternative to convolutional neural networks. This new architecture is used successfully to get state of the art performance for a number of common image generation benchmarks when compared with non-autoregressive approach (even much larger CNNs). There is a lot of useful feedback in the reviews themselves: a thing to consider in the final version is the fact that the authors had motivated SDNs as drop-in replacements for CNNs, but do experiments mostly in VAE-like settings. This is a point that was raised by multiple reviewers and is clearly something that should be dealt with as explicitly as possible.

While there are legitimate reasons to be wary of the increased computation time, I tend to side with the authors that baselines that are being compared with SDNs are likely to have more optimized primitives. From the inference numbers presented in the rebuttal, it doesn't appear like the speed issues are insurmountable.

Given the high quality of writing, the excellent performance on image density modeling, the various ablations and understanding of the disentangling effects, I think this is an interesting piece of work that the field would benefit from.